# Inkjet Printing of High Aspect Ratio Silver Lines via Laser-Induced Selective Surface Wetting Technique

Iseok Sim [1,2], Seongju Park [1], Kwon-Yong Shin [1], Chanwoo Yang [1], Heuiseok Kang [1], Jun Young Hwang [1,*] and Seung-Jae Moon [2]

1    Korea Institute of Industrial Technology, Cheonan 31056, Republic of Korea
2    Department of Mechanical Engineering, Hanyang University, Seoul 04763, Republic of Korea
*    Correspondence: jyhwang@kitech.re.kr

**Abstract:** The field of printed electronics for highly integrated circuits and energy devices demands very fine and highly conductive electric interconnections. In this study, conductive lines having a high cross-sectional aspect ratio were printed via the inkjet printing of Ag nanoparticle inks assisted by a laser-induced selective surface wetting technique: a hydrophobic layer of self-assembled monolayer-treated ZnO nanorods was coated on a glass substrate and selectively ablated by a laser to form micro-channels for the inkjet, whose surface energy changed from 36.3 mJ/m$^2$ to 51.5 mJ/m$^2$ before and after the laser irradiation. With the varying width of the laser-ablated channels and pitch of jetted ink drops, the 3D shapes of the printed silver lines were measured to investigate their effects on the widths, heights, and uniformities of the printed patterns. The results showed that the present technique realized a uniform line of 35 μm width and 0.46 μm average thickness, having an aspect ratio of 0.013, which is 7.6 times higher than that printed on bare glass.

**Keywords:** inkjet printing; super-hydrophobic coating; selective surface wetting; laser ablation; Ag nanoparticle ink; conductive lines; high aspect ratio





## 1. Introduction

The printed electronics industry has seen significant growth and advancements, driven by the increasing demand for low-cost, flexible, large-area, lightweight electronic devices. In highly integrated electronics and opto-electrical energy devices, metallic patterns having high conductivity and high resolution are essential for electrical electrodes and interconnections, such as antennae for internet of things (IoT) devices [1,2], electrodes for micro-sensors and thin film transistors [3–5], electrodes for micro-supercapacitors [6,7], fingers for solar cells [8,9], and grid electrodes for touch sensors and organic light-emitting diode (OLED) lighting [10,11].

Among the various printing techniques, inkjet plays an irreplaceable role as a digital additive manufacturing technology. Since inkjet printing was first introduced to fabricate metallic micro-lines using Ag nanoparticle inks [12], a number of studies have been conducted not only to realize the printing of narrow and uniform lines [13–16] but also to achieve the high conductivity of the printed pattern by improving the ink formulation and thermal treatment [17–19].

However, the printed metallic patterns do not have electrical conductivity that is as high as those produced by etching or plating because the sintered or reduced metals from inks have porous nanostructures [18,19]. Moreover, the aspect ratios of inkjet-printed patterns on homogeneous substrates are very low, less than 0.05, due to the limitations of low metallic content in the inks and a low contact angle between the inks and the substrates, which are limited by the jetting viscosity and printing stability, respectively [15,16,20]. As a result, the reduced electrical conductivity and the low aspect ratio have led to a dilemma in which the electrical resistance of an inkjet-printed metallic pattern is significantly increased with its resolution.

In this study, uniform silver lines with high aspect ratios were printed using laser-selective surface treatment and inkjet printing. Ultraviolet (UV) absorptive super-hydrophobic layers of self-assembled monolayer (SAM)-treated ZnO nanorods were coated on glass substrates, which were selectively ablated by a nano-second UV pulse laser to form micro-channels having various widths. Ag nanoparticle ink drops were ejected on the channels using a piezoelectric inkjet print head to print narrow silver lines having various cross-sectional shapes. With the varying widths of the laser-ablated channels and pitches of ink drops jetted on the channels, the topological shapes of the printed silver lines were measured to investigate their effects on the widths, heights, and uniformities of the printed patterns.

## 2. Materials and Methods

As shown in Figure 1, the present process consists of hydrophobic layer coating, laser-selective surface treatment, and inkjet printing, in which the coating process includes seed layer coating, ZnO nanorod growth, and self-assembled hydrophobic monolayer formation. A soda–lime slide glass (Marienfeld, 1000412, Berlin, Germany) was used as a substrate. Substrates were cleaned with isopropyl alcohol, acetone, and deionized water sonication and treated with UV ozone for 10 min. After cleaning, a 10 mM zinc acetate dihydrate (99.999%, Sigma-Aldrich Inc., Seoul) solution was spin-coated onto the cleaned substrates at 2000 rpm for 30 s. Then, the samples were baked at 130 °C for 1 min. We repeated the above coating and baking process five times to ensure uniform coverage of zinc acetate crystallites. All samples were then pre-annealed in an air atmosphere at 200 °C for 1 h to yield ZnO seeds. For ZnO nanorod growth, samples were soaked in 25 mM zinc nitrate hexahydrate (98%, Sigma-Aldrich) and 25 mM hexamethylenetetramine (99%+, Sigma-Aldrich, St. Louis, MO, USA) in a deionized water solution, followed by heating in a microwave oven (2.45 GHz) at 1000 W for 3 min. To remove the residual solvent, the samples were rinsed with deionized water [21]. To form the SAM of the 1H,1H,2H,2H-perfluorooctyltriethoxysilane (PFOTES, 98%, Sigma-Aldrich) on the ZnO nanorods, the samples were dipped into a 1 mM PFOTES and 0.01 vol.% butylamine in toluene solution, at 45 °C for 90 min. The treated samples were rinsed with toluene to remove excess PFOTES molecules [22]. Finally, to expose the hydrophilic surfaces of micro-channels having various widths, these super-hydrophobic samples were selectively ablated by a nano-second UV pulse laser (HGTECH, LSU5DS 3D Laser Marking System, Wuhan, China). The widths of the ablated channels were 20, 40, 60, 80, and 100 μm. The wavelength of the laser was 355 nm, the pulse frequency was 100,000 Hz, the off-time was 6500 ns, and the scan rate was 1000 mm/s.

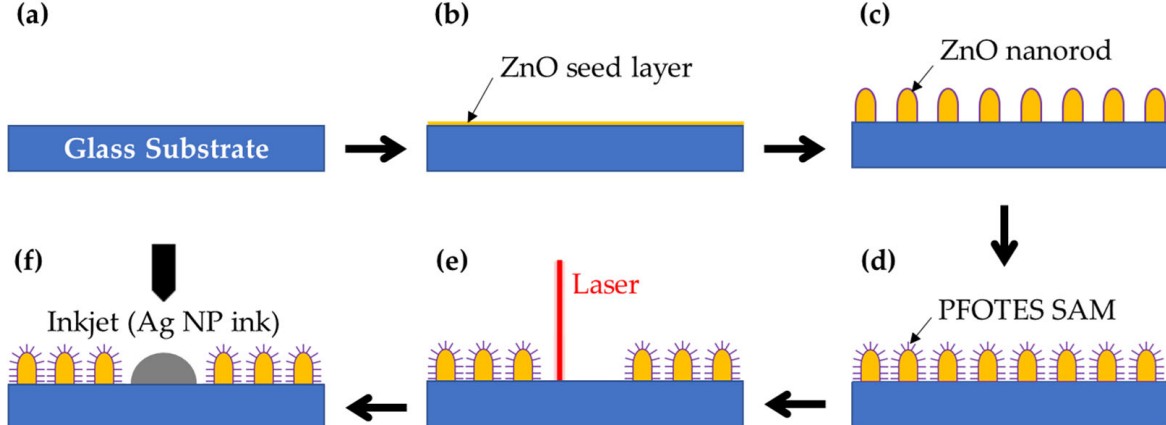

**Figure 1.** Schematic of the present inkjet process via laser-induced selective surface wetting technique: (**a**) substrate cleaning, (**b**) seed layer coating, (**c**) ZnO nanorod growth, (**d**) formation of PFOTES SAM, (**e**) laser surface treatment, and (**f**) inkjet printing of Ag NP.

Figure 2 and Table 1 show the changes in the nanostructure and wetting characteristics of the ZnO-PFOTES coating after the laser treatment, measured by field-emission scanning electron microscopy (FE-SEM, FEI Sirion, Hillsboro, OR, USA) and a drop shape analyzer (DSA100, KRÜSS), respectively. In Figure 2b, the boundary between the laser-treated and untreated zones is also shown. The energy profile of the UV laser beam used in this study has a Gaussian distribution, so the boundary region in Figure 2b, where the edge of the relatively low-energy laser beam was hit, resulted in irregularly sized pores where the ZnO-PFOTES coating did not collapse. To quantitatively show the difference in the wettability of ZnO-PFOTES-coated surfaces due to laser treatment, the surface energy of the coated surfaces was investigated by measuring the contact angles of two test liquids, water and diiodomethane. The surface energy is a direct indicator of intermolecular forces and is composed of the sum of the dispersive and polar components and is calculated by the following equation [23]:

$$1 + \cos\theta = \frac{2\left(\gamma_s^d\right)^{1/2}\left(\gamma_{lv}^d\right)^{1/2}}{\gamma_{lv}} + \frac{2\left(\gamma_s^p\right)^{1/2}\left(\gamma_{lv}^p\right)^{1/2}}{\gamma_{lv}}$$

where $\gamma_s$ and $\gamma_{lv}$ are the surface energies of the sample and test liquid, respectively, and the superscripts d and *p* refer to dispersion and polar components, respectively. In addition, the values of $\gamma_{lv}$, $\gamma_{lv}^d$, and $\gamma_{lv}^p$ for the test liquids and the procedure for solving the equation are provided in reference [23]. Laser irradiation collapses the SAM structure of the ZnO-PFOTES coating, resulting in the increase in surface energy, $\gamma_s$, from 36.3 mJ/m$^2$ to 51.5 mJ/m$^2$. Interestingly, as summarized in Table 1, the polarity ratio, $X_p$, of the surface energy is greatly increased by laser treatment because the polar component, $\gamma_s^p$, increases from 0.513 mJ/m$^2$ to 25.9 mJ/m$^2$ but the dispersive component, $\gamma_s^d$, decreases from 35.8 mJ/m$^2$ to 25.6 mJ/m$^2$. As a result, the contact angle of triethylene glycol monoethyl ether (TGME), which is the main solvent of the ink, changes from 40.4° to 5.6° with the laser treatment. In Figure 2b, the boundary between the laser-treated and untreated zones is shown.

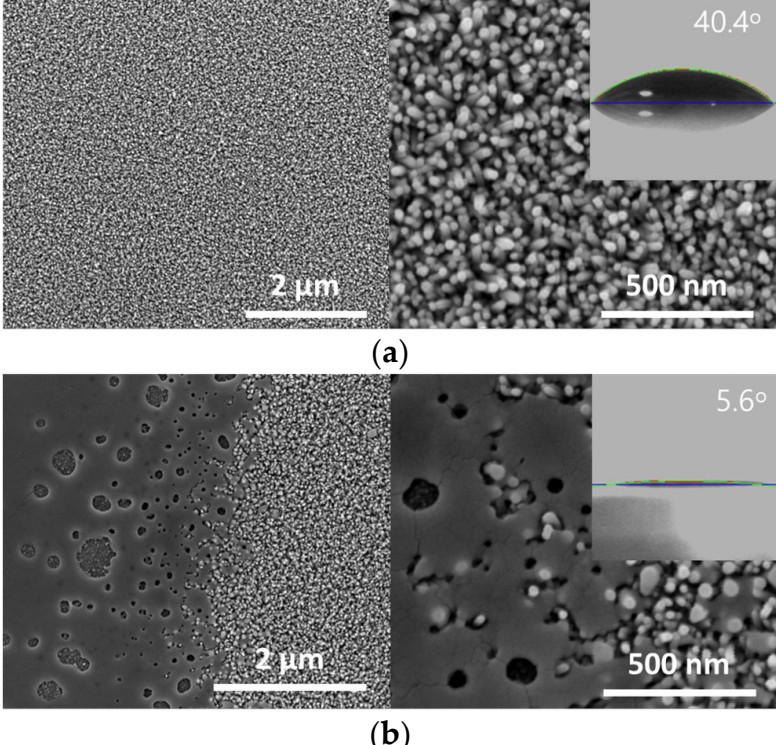

**Figure 2.** FE-SEM top-view images of surface nano-morphologies of ZnO-PEOTES coatings and contact angle images of TGME on the coatings: (**a**) before laser treatment, (**b**) after laser treatment.

**Table 1.** Surface energy characteristics of pristine and laser-treated ZnO-PEOTES coatings.

| | Contact Angle (Degree) | | | Surface Energy (mJ m$^{-2}$) | | | $X_p(\gamma_s^p/\gamma_s)$ |
|---|---|---|---|---|---|---|---|
| | Deionized Water | Diiodomethane | TGME | $\gamma_s^p$ | $\gamma_s^d$ | $\gamma_s(\gamma_s^p+\gamma_s^d)$ | |
| Pristine ZnO-PFOTES | 111.3 | 53.3 | 40.4 | 0.513 | 35.8 | 36.3 | 0.0141 |
| Laser-treated ZnO-PFOTES | 49.0 | 46.3 | 5.6 | 25.9 | 25.6 | 51.5 | 0.503 |

Silver lines were printed on the laser-ablated channel with various drop spacings. The ink used in this study was a commercial silver nanoparticle ink (ANP, DGP 40LT-15C, Sigma Aldrich, St. Louis, MO, USA), which contained 30~35 wt% of spherical silver nanoparticles having diameters smaller than 50 nm, dispersed in TGME. The density, viscosity, and surface tension of the ink were 1.44 g/mL, 12.8 cP, and 35.9 dyne/cm, respectively. The custom-made drop-on-demand (DOD) inkjet printing equipment consisted of a piezoelectric jetting head and a 2D traverse stage. The traverse system was configured to cover a 300 mm × 300 mm printing area with 1 μm movement accuracy. The jetting head (Dimatix, Samba, West Yorkshire, UK) had 12 nozzles with a diameter of 30 μm and generated a series of droplets at the jetting frequency of 1 kHz. The driving signal for piezoelectric deformation had a single trapezoidal waveform with a rise time, dwell time, and fall time of 1.0 μs, 1.2 μs, and 1.0 μs, respectively, and an amplitude of 32 V. Falling droplets had an average diameter of 17.4 μm and a volume of 2.7 pl, as shown in Figure 3. The three-dimensional shapes of the printed lines were measured with a 3D profiler (NANO SYSTEM, NV-F2700, Daejeon, Korea), which had a measurement resolution of 0.5 nm along the Z axis and 0.64 μm for the X–Y axis. The average width and thickness of a printed line were obtained from the measured profiles of five cross-sections at arbitrary locations along the line.

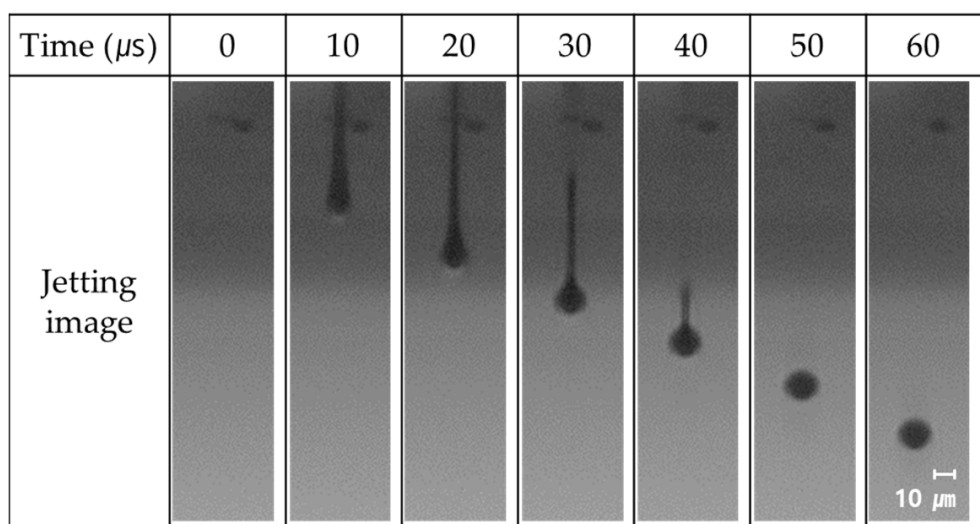

**Figure 3.** Sequential images of droplet jetting from inkjet nozzle.

### 3. Results

Figure 4 shows the 2D and 3D profiles of the printed silver lines on the surfaces with and without the present selective surface wetting technique. Here, the pitch of the jetted drops is 5 μm along the lines, which is very small compared to the drop diameter of 17.4 μm. To print a line having a large thickness, a small pitch of jetted drops is needed. However, when a line is printed on a homogeneous substrate with such a small drop pitch, fluid-dynamic instabilities in the printed ink bead result in a non-uniform profile of the line, having bulges [15,16], which appear in the printed line on the bare glass in Figure 4a.

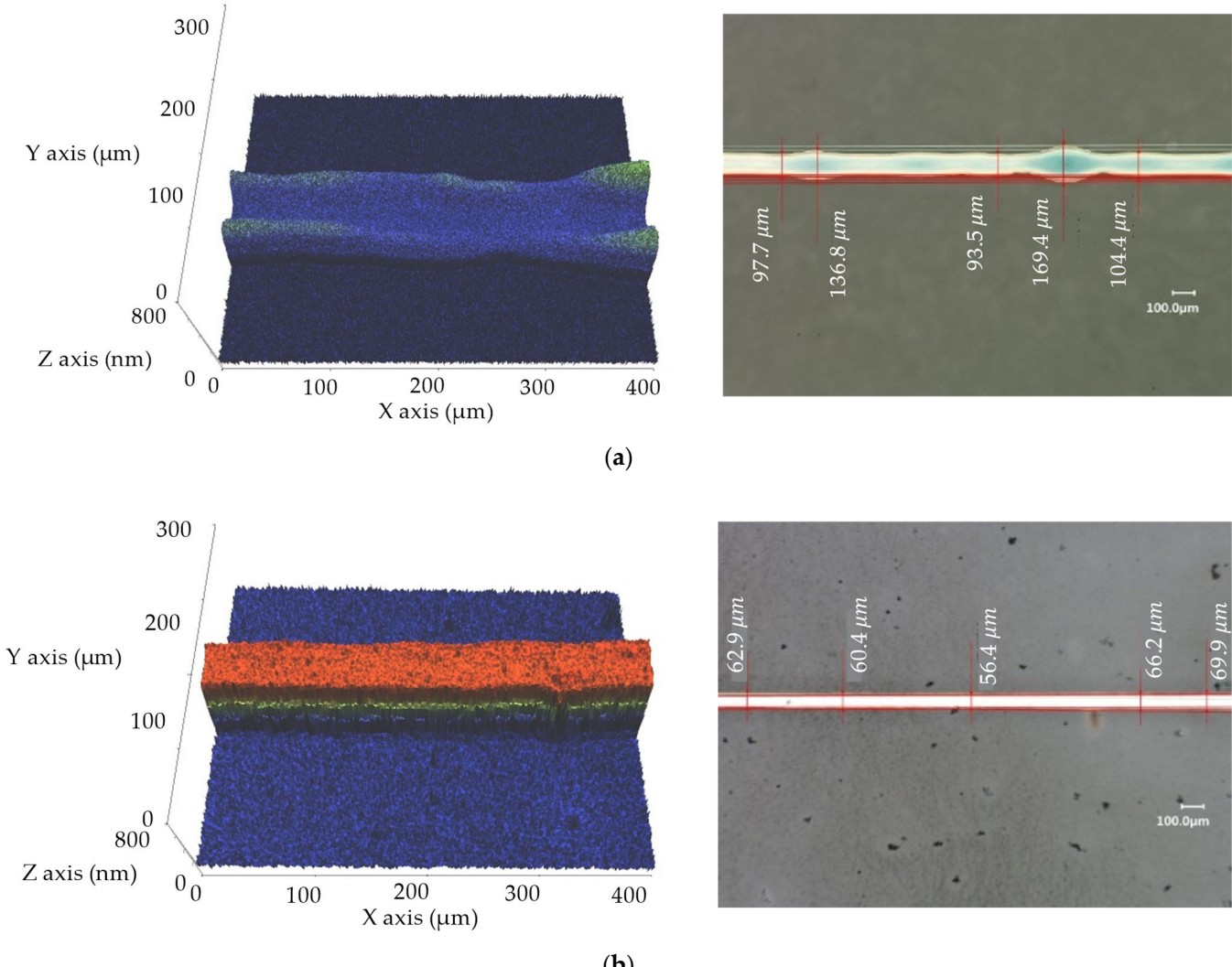

**Figure 4.** The 3D and 2D profiles of printed silver lines on (**a**) bare glass and (**b**) the surface with selective wetting technique: inkjet drop spacing is 5 μm for both cases and laser ablation width is 60 μm for case (**b**).

When the selective surface wetting technique is introduced, as shown in Figure 4b, the profile of the printed pattern is dramatically changed. Compared with the pattern on the bare glass, the width becomes smaller and the thickness becomes larger. Moreover, the uniformities of both width and thickness are greatly increased. Encouraged by these results, a more detailed study was conducted to understand the effects of the inkjet drop spacing and laser ablation width on the printing of narrow lines that have a high aspect ratio and high uniformity.

Figure 5 demonstrates the effect of the laser-ablated channel width of the selectively wetted surface, showing 2D and 3D profiles of printed lines with a drop spacing of 10 μm and with various laser ablation widths of 80, 40, and 20 μm. When the channel width is relatively large at 80 μm, unevenness in the width and thickness remains even if the generation of bulges disappears. As the channel width decreases to 40 μm, the thickness of the pattern increases and the uniformity of the thickness and width is improved. If the channel width further decreases to 20 μm, the ink in the channel floods over the surface energy barrier so that bulges are generated intermittently.

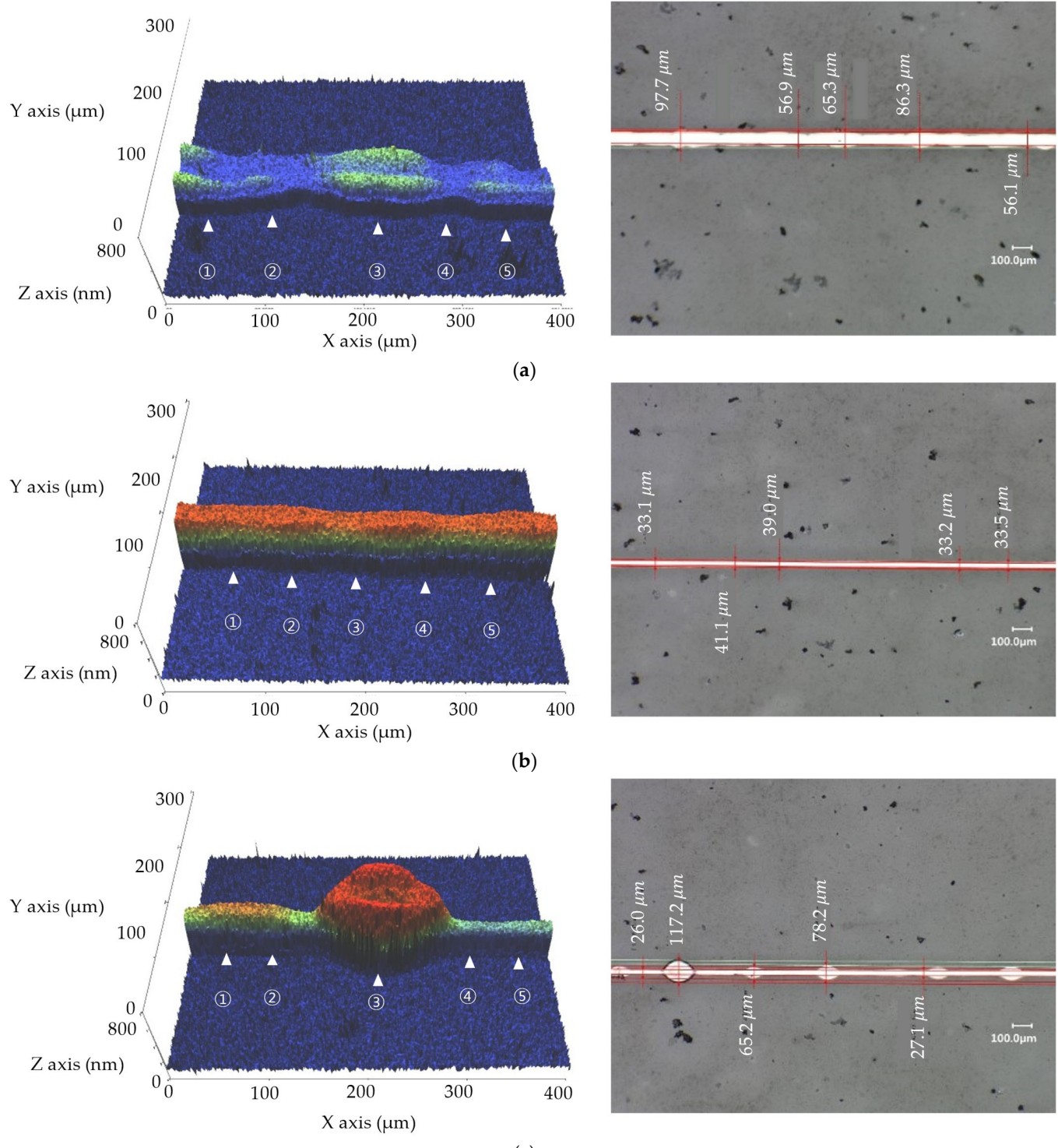

**Figure 5.** The 3D and 2D profiles of printed silver lines with inkjet drop spacing of 10 μm and various laser ablation widths: (**a**) 80 μm, (**b**) 40 μm, and (**c**) 20 μm.

Figure 6 shows the effect of the inkjet drop spacing on the shape of the pattern printed on the selectively wetted surface having a channel width of 40 μm. As the drop pitch decreases from 20 μm to 5 μm, the thickness of the printed lines naturally increases. However, when the drop pitch becomes 5 μm, bulge formation occurs again because an excessive volume of ink is supplied in the unit length of the channel. On the contrary, for the higher drop pitch of 20 μm, the 3D profile of the pattern shows a neck in which the

cross-sectional area decreases sharply. The results in Figures 3 and 4 imply that there is an optimal channel width for a uniform pattern as a function of the drop pitch.

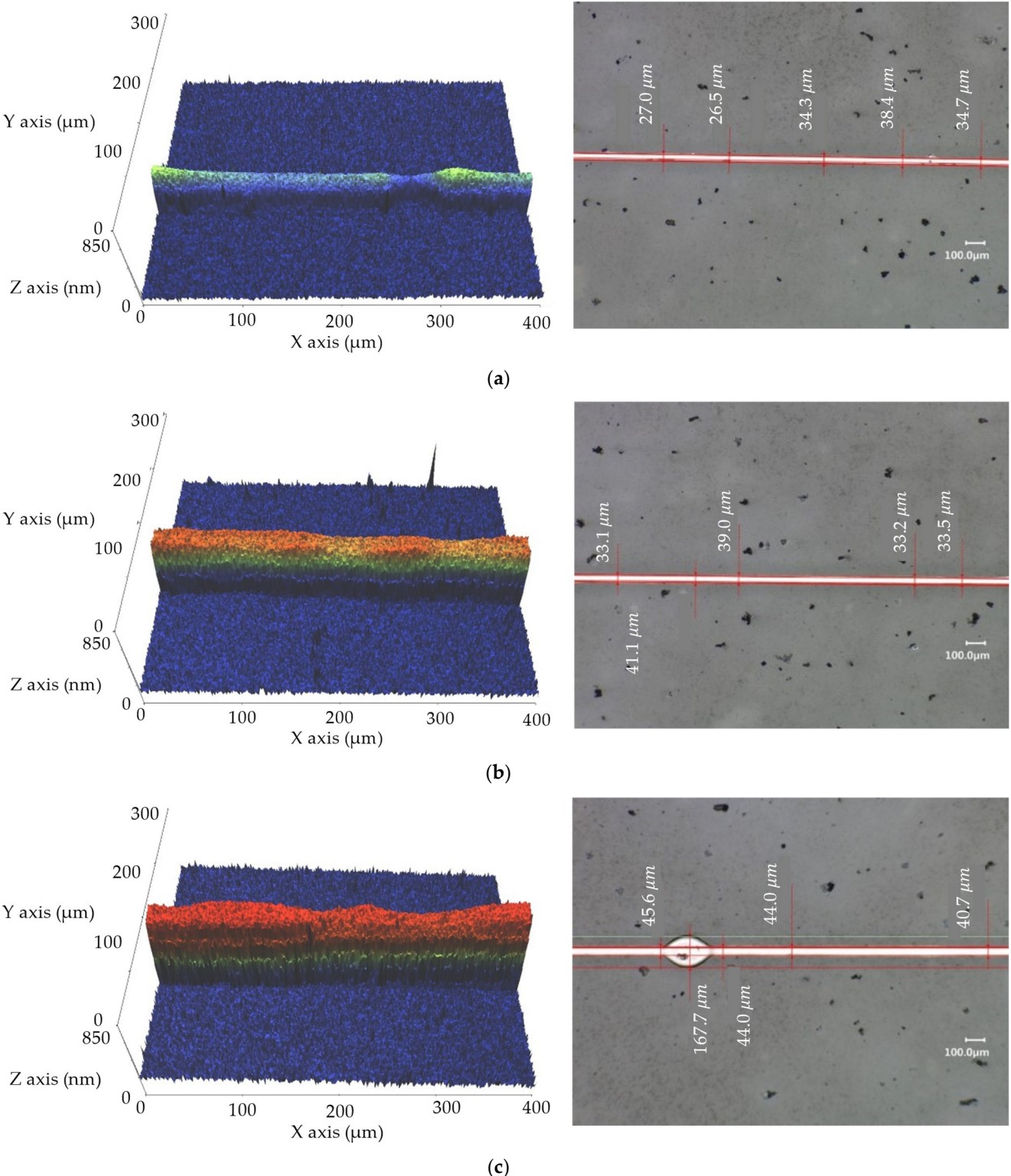

**Figure 6.** The 3D and 2D profiles of printed silver lines with laser ablation width of 40 μm and various inkjet drop spacings: (**a**) 20 μm, (**b**) 10 μm, and (**c**) 5 μm.

In order to investigate the uniformity of the printed pattern along the lines, cross-sectional distributions of thickness at five arbitrary locations of each line are shown in Figure 7, where the printing conditions are the same as those in Figure 5. It is clearly seen that, among the three laser-ablated channel widths of 80 µm, 40 µm, and 20 µm, the cross-sectional profiles are most uniform for the channel width of 40 µm, where the full width half maximum (FWHM) ranges from 38.8 µm to 41.5 µm, showing single modal distributions having an average thickness of 0.458 µm. When the width of the laser-ablated channel is wider at 80 µm, the deviation between distributions increases, so that the FWHM varies from 44.9 µm to 70.5 µm. When the channel width is narrower at 20 µm, intermittent bulging occurs, as mentioned in Figure 5. Then, thicker and wider distributions are generated at the bulges, while the distributions become thinner for the necks adjacent to the bulges. As a result, the FWHM varies greatly from 20.0 µm to 79.7 µm near the bulge, even though the profiles located far away from the bulges are relatively uniform.

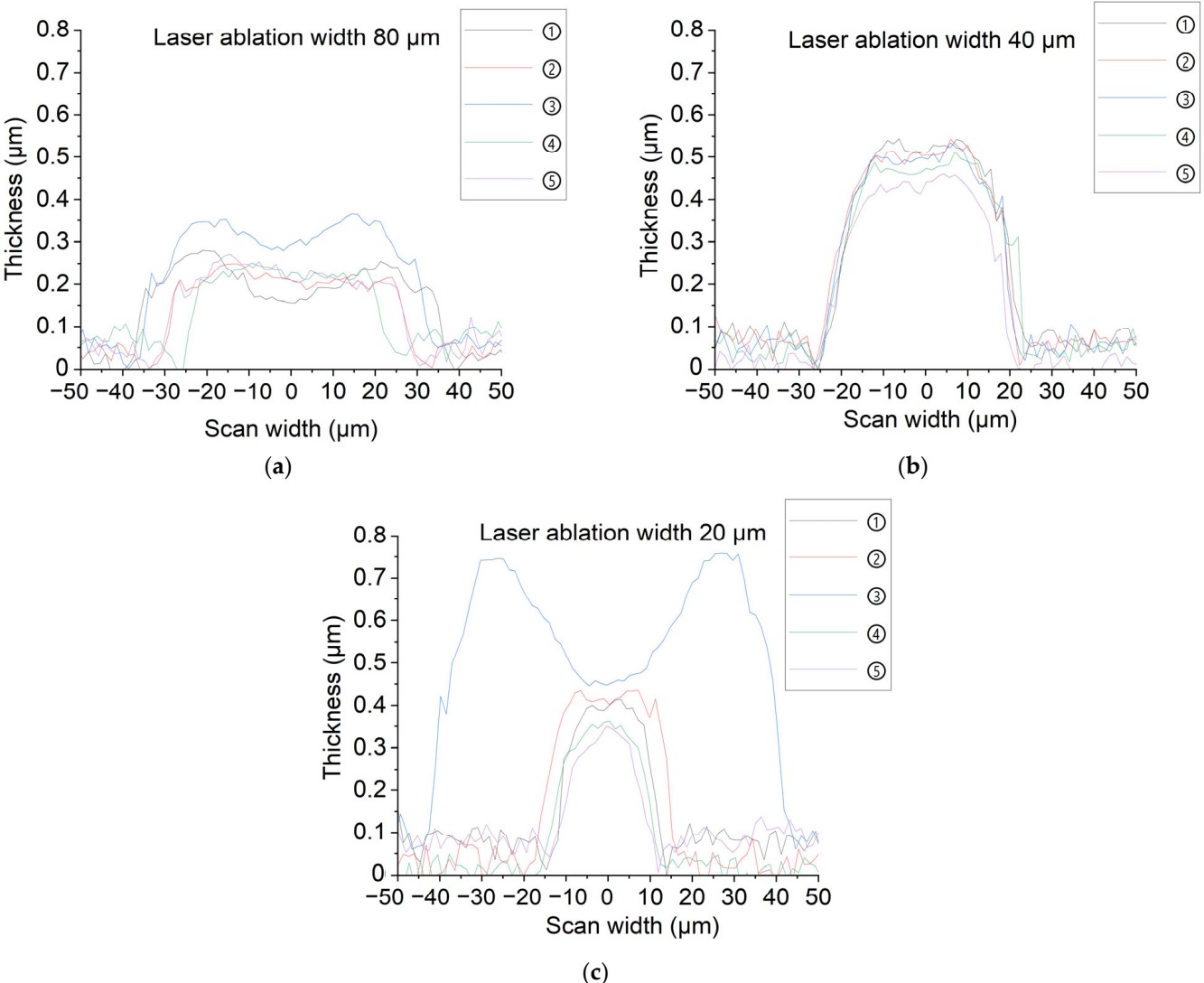

**Figure 7.** Cross-sectional thickness profiles at five arbitrary locations (marked in Figure 5) along the printed lines with inkjet drop spacing of 10 µm and various laser ablation widths: (**a**) 80 µm, (**b**) 40 µm, and (**c**) 20 µm.

Figure 8 shows the overall changing aspects of the line width and thickness with the laser ablation width of the hydrophobic coating and the inkjet drop spacing. It is generally seen that, with the decrease in the laser ablation width, the line width decreases and the thickness increases, and the deviations in both the width and the thickness decrease. However, if the channel width decreases excessively so that bulges are generated, then the deviations in the width and the thickness begin to increase. The increase in the line thickness is also limited when massive bulges are generated because they absorb nearby ink. The critical channel width that had the smallest value without generating a bulge was 60 μm for the droplet spacing of 5 μm, which decreased to 40 μm for the larger droplet spacings of 10 μm and 20 μm. As a result, most high cross-sectional aspect ratios could be obtained at the critical channel width for each drop spacing, as shown in Figure 9. Specifically, in the present experiments, the maximum aspect ratio of the printed line was 0.013 at a drop spacing of 10 μm and a channel width of 40 μm, which is 7.6 times higher than that printed on bare glass.

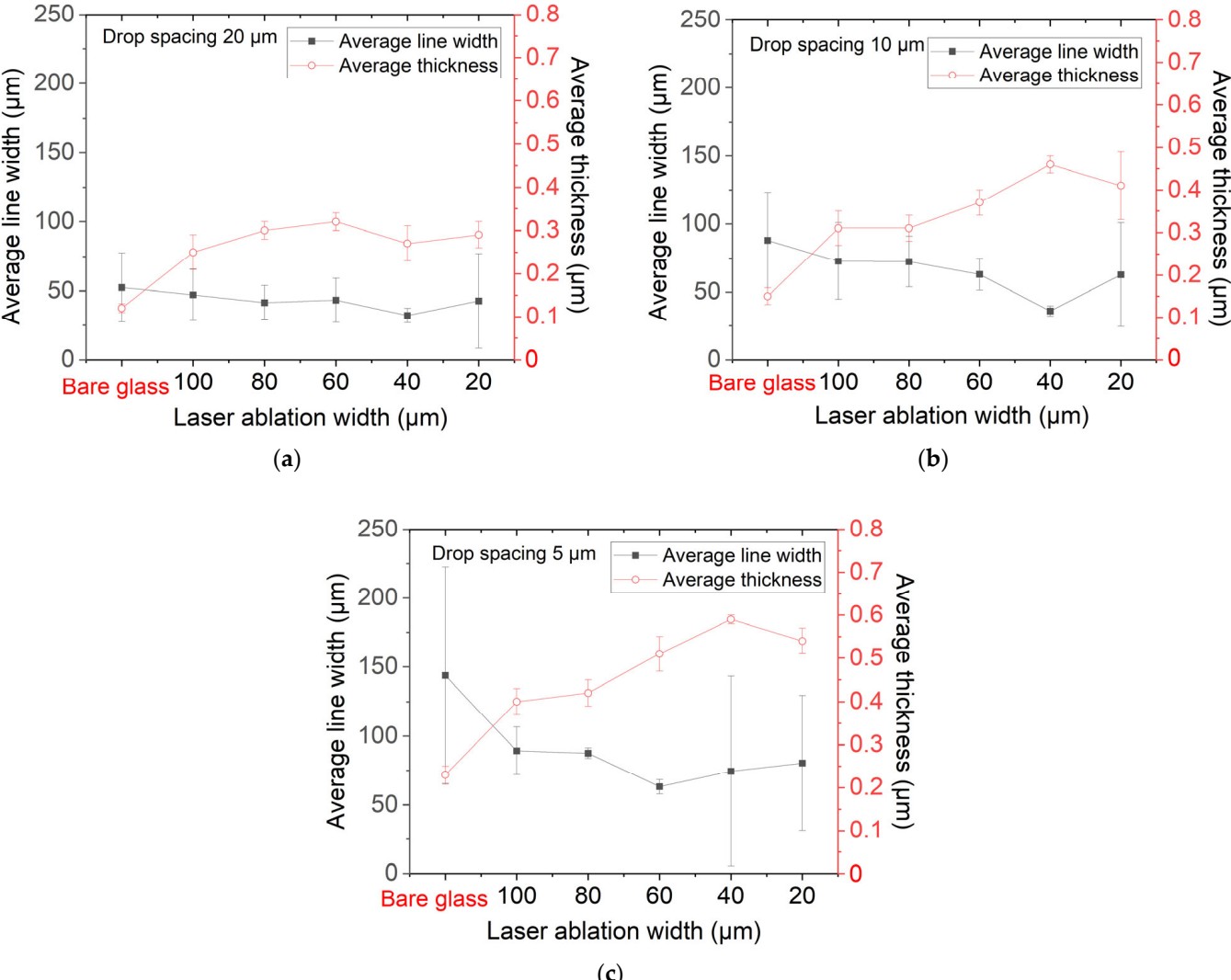

**Figure 8.** Variation in printed line width and thickness as a function of laser ablation width for inkjet drop spacing of (**a**) 20 μm, (**b**) 10 μm, and (**c**) 5 μm.

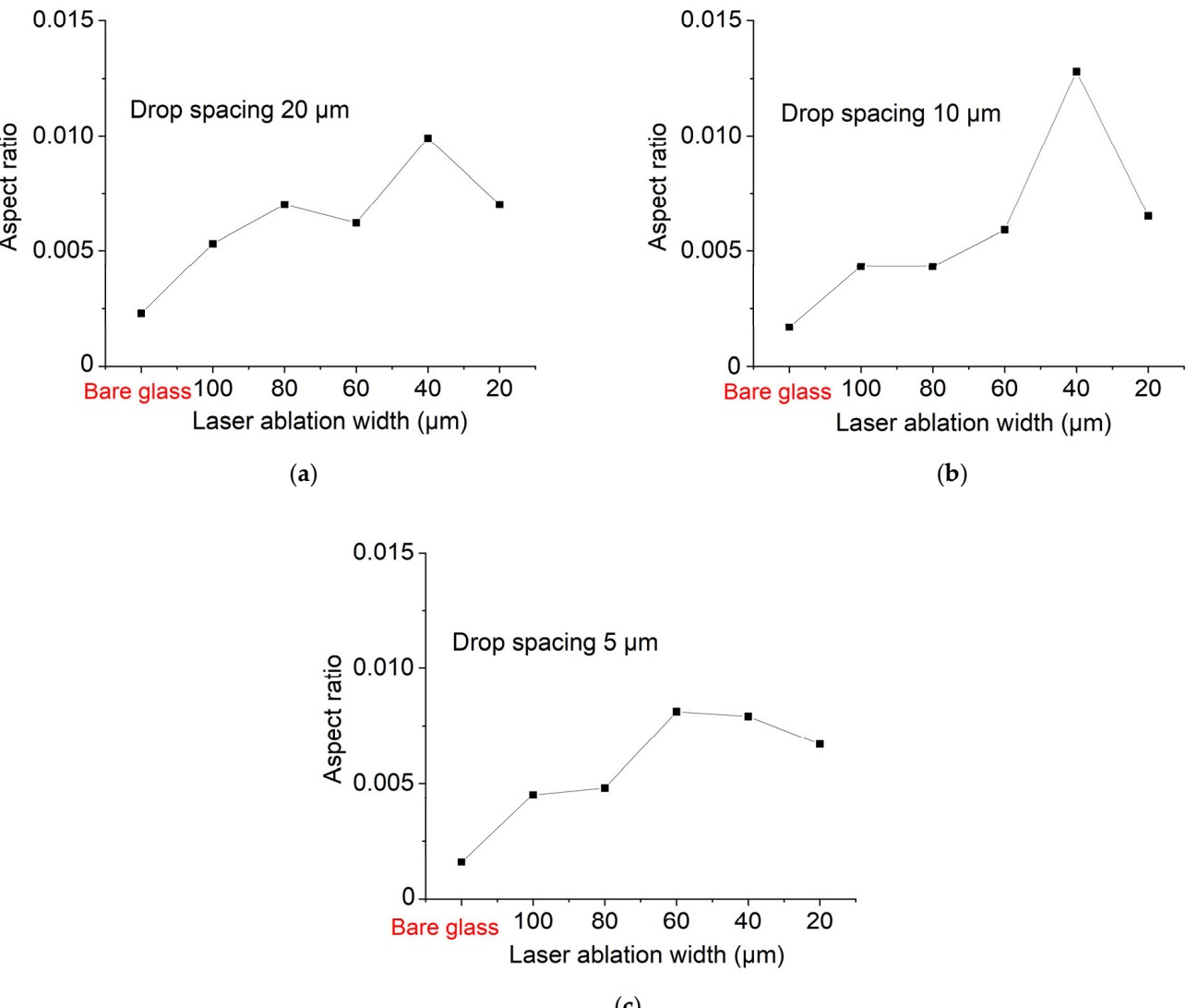

**Figure 9.** Variation in cross-sectional aspect ratio of printed lines as a function of laser ablation width for inkjet drop spacings of (**a**) 20 μm, (**b**) 10 μm, and (**c**) 5 μm.

## 4. Conclusions

High aspect ratio silver lines were successfully fabricated on glass substrates via the inkjet printing of Ag nanoparticle ink, assisted by a laser-induced selective surface wetting technique. Self-assembled hydrophobic monolayer-treated ZnO nanorods were coated on glass substrates to have surface energy of 36.3 mJ/m$^2$, which were selectively ablated by a nano-second UV pulse laser to form micro-channels, increasing the surface energy selectively to 51.5 mJ/m$^2$. The effect of the surface energy barrier became stronger when the channel width was narrower and the jetted drop volume per the channel length was larger. However, there existed a critical channel width, having the smallest value without generating a bulge, because ink flooded over the energy barrier to form bulges when the channel width decreased excessively. As a result, the present study obtained the highest cross-sectional aspect ratio of 0.013 at a critical channel width of 40 μm and drop spacing of 10 μm, achieving an average thickness of 0.46 μm and an average line width of 36.0 μm with a single print.

**Author Contributions:** Conceptualization, J.Y.H. and C.Y.; methodology, I.S., S.P. and K.-Y.S.; valida-tion, K.-Y.S. and C.Y.; formal analysis, I.S. and S.P.; investigation, I.S., C.Y. and J.Y.H.; writing—original draft preparation, J.Y.H., I.S. and C.Y.; writing—review and editing, C.Y. and J.Y.H.; visualization, I.S.; supervision, H.K., J.Y.H. and S.-J.M. All authors have read and agreed to the published version of the manuscript.

**Funding:** This research was funded by the Korea Institute of Industrial Technology as "Development of root technology for multi-product flexible production" (KITECH EO-23-0008).

**Conflicts of Interest:** The authors declare no conflict of interest.

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
