# Peer review of "Inkjet Printing of High Aspect Ratio Silver Lines via Laser-Induced Selective Surface Wetting Technique"

_coatings, doi:10.3390/coatings13040683_

Round 1

Reviewer 1 Report

The paper is well-written and well-structured, providing a clear explanation of the methodology and results. The authors present original research and data, which demonstrate the effectiveness of the laser-induced selective surface wetting technique for inkjet printing high aspect ratio silver lines. The results show that the technique can produce lines with aspect ratios of up to 0.013 which is 7.6 times higher than that printed on a bare glass. This is a significant improvement over traditional inkjet printing methods. The authors also provide a thorough analysis of the underlying mechanisms that drive the laser-induced selective surface wetting process. This new method has significant potential for use in various fields, including electronics and biomedicine. Based on the review, I recommend this paper for publication in the journal. The paper's potential impact in various fields makes it a valuable contribution to the literature.

Author Response

Thank you very much.

Reviewer 2 Report

The author has conducted a comprehensive study of the high aspect ratio silver lines on glass substrates via inkjet printing with laser-induced selective surface-wetting techniques. The effects of the width of the laser and shapes of the printed silver lines are fully investigated on the widths, heights, and uniformities of the printed patterns. The characterizations of SEM, surface energy calculation, droplet jetting method, and AFM are used and the results are fully defined and deeply discussed. It is qualified for publication but some minor comments were given below:

  1. The differences caused by laser treatment in Figs. 2(a) and (b) can be discussed in more detail; Also what’s the reason to cause the pores in Fig. 2(b) and why the pore size is not uniform?

  2. The text in Fig. 4 can be more clear; The same for Figs. 5-6.

  3. For Fig. 7, please explain the different colors of lines in each image. Lacking a figure legend?

Author Response

The authors thank you for your careful comments.

  1. The differences caused by laser treatment in Figs. 2(a) and (b) can be discussed in more detail; Also what’s the reason to cause the pores in Fig. 2(b) and why the pore size is not uniform?

>> We appreciate the reviewer’s comment. The energy profile of the UV laser beam used in this study has a Gaussian distribution, so the boundary region in Figure 2(b), where the edge of the relatively low energy laser beam was hit, resulted in irregularly sized pores where the ZnO-PFOTES coating did not collapse.

According to your valuable suggestion, we added a further explanation to in the second paragraphs of section 2 of the revised manuscript:

“The energy profile of the UV laser beam used in this study has a Gaussian distribution, so the boundary region in Figure 2(b), where the edge of the relatively low energy laser beam was hit, resulted in irregularly sized pores where the ZnO-PFOTES coating did not collapse.”

  1. The text in Fig. 4 can be more clear; The same for Figs. 5-6.

>> The texts in the figures are enlarged.

  1. For Fig. 7, please explain the different colors of lines in each image. Lacking a figure legend?

>> Figure legends are added in Fig. 7.

Reviewer 3 Report

This work presents the fabrication of high aspect-ratio conductive lines with inkjet printing after the surface energy modification with laser ablation. I have questions below.

1. The introduction part needs to be improved. What's the achieved aspect ratio with existing techniques in literature? What' the main contribution of the current work? 

2. Why ZnO2 nanorods layer? Does it limit the application scope of the proposed fabrication scheme?

3. The electrical resistance of the printed conductive lines needs to be studied.

4. Table 2, how are the polar component and dispersive component calculated?

5. Page 4 line 127, Figure 4a?

6. The comparison with other fabrication techniques are necessary. 

Author Response

The authors thank you for your valuable comments.

  1. The introduction part needs to be improved. What's the achieved aspect ratio with existing techniques in literature? What' the main contribution of the current work?

>> For a substrate having homogeneous surface energy, the contact angle between ink and the substrate is limited by printing stability. In addition, metallic content in an ink is also limited to a few tends of percent in weight or a few percent in volume due to viscosity affordable for jetting. As a result, typical aspect ratio of a printed metal nanoparticle line is less than 0.05 [16. 20]. In this study, it was shown that the aspect ratio could be increased without printing stability problem using selective surface treatment. This point was added to the introduction part of the revised manuscript:

“Moreover, the aspect ratios of inkjet-printed patterns on homogeneous substrates are very low, less than 0.05, due to the limitations of low metallic content in the inks and low contact angle between the inks and the substrates, which are limited by jetting viscosity and printing stability, respectively [15,16,20].”

  1. Why ZnO2 nanorods layer? Does it limit the application scope of the proposed fabrication scheme?

>> ZnO has the property of absorbing UV light well, so it is suitable for implementing the selective surface wetting technique using UV laser ablation in this study. In addition, ZnO is adopted in this study because it has the advantages of low cost and simple process. In particular, ZnO nanorods can be easily produced in a large area by solution process at low temperature (100 ℃), so the proposed fabrication method is expected to have no limitations in practical application.

  1. The electrical resistance of the printed conductive lines needs to be studied.

>> The authors are fully agreed with the reviewer that the electrical resistance of the printed conductive lines can be affected by pattern morphology [18], which is very important to a practical application. However, the authors think that the present study on the printing itself is also worth reporting. An investigation of electrical resistance as a function of aspect ratio would be our next study.

  1. Table 2, how are the polar component and dispersive component calculated?

>> According to your valuable suggestion, we added a further explanation to in the second paragraphs of section 2 of the revised manuscript:

“To quantitatively show the difference in wettability of ZnO-PFOTES-coated surfaces due to laser treatment, the surface energy of the coated surfaces was investigated by measuring the contact angles of two test liquids, water and diiodomethane. The surface energy is a direct indicator of intermolecular forces and is composed of the sum of the dispersive and polar components and is calculated by the following equation [23]:

<Equation>

where γ_s and γ_lv are the surface energies of the sample and test liquid, respectively, and the superscripts d and p refer to dispersion and polar components, respectively. In addition, the values of γ_lv, γ_s^d, and γ_s^p for the test liquids and the procedure for solving the equation are provided in reference [23].”

Also, we added the following a related reference:

  1. Kinloch A.J., Adhesion and Adhesives: Science and Technology, Chapman and Hall, 1987, Chap. 2, 18–32.

  1. Page 4 line 127, Figure 4a?

>> This was cleared in the revised manuscript.

  1. The comparison with other fabrication techniques are necessary.

>> Inkjet printing of uniform narrow line is a subject that many researchers have been interested in and studied. The authors believe that this study has clearly shown that the present selective surface treatment technique can be successfully adopted for morphological control of inkjet-printed pattern.

Round 2

Reviewer 3 Report

The achieved aspect ratio is less than 0.05 in literature. This work reports an aspect ratio of 0.013. This doesn't seem to be an improvement, at least in terms of aspect ratio.

Other than that, my questions have been well addressed.